# Antibiotic Use in Pediatrics: Perceptions and Practices of Romanian Physicians

**DOI:** 10.3390/antibiotics14100976

**Published:** 2025-09-27

**Authors:** Alin Iuhas, Radu Galiș, Marius Rus, Codruța Diana Petcheși, Andreea Balmoș, Cristian Marinău, Larisa Niulaș, Zsolt Futaki, Dorina Matioc, Cristian Sava

**Affiliations:** 1Department of Medical Disciplines, Faculty of Medicine and Pharmacy, University of Oradea, 410073 Oradea, Romania; 2Pediatrics Department, Bihor County Clinical Emergency Hospital, 410167 Oradea, Romania; 3Neonatology Department, Bihor County Clinical Emergency Hospital, 410167 Oradea, Romania; 4Cardiology Department, Bihor County Clinical Emergency Hospital, 410167 Oradea, Romania; 5Department of Preclinical Disciplines, Faculty of Medicine and Pharmacy, University of Oradea, 410073 Oradea, Romania; 6Regional Center of Medical Genetics Bihor, Bihor County Clinical Emergency Hospital (Part of ERN ITHACA), 410053 Oradea, Romania

**Keywords:** antibiotic resistance, antibiotic consumption, antimicrobial stewardship, children, physicians, Romania

## Abstract

**Background/Objectives**: The global threat of antimicrobial resistance is a significant public health challenge, leading to prolonged hospitalizations, increased costs, and elevated mortality. Romania faces one of Europe’s highest burdens of antimicrobial consumption and resistance. This study aimed to investigate the factors that influence antibiotic prescribing practices among physicians in pediatric care in Romania. **Method**: This quantitative, cross-sectional study collected data using a self-administered, structured questionnaire from 154 healthcare professionals (family physicians, pediatricians, and other specialists) providing pediatric care in Romania. Participants were recruited via non-probability convenience sampling. The 29-question survey gathered demographic data and explored perceptions and practices regarding antibiotic therapy in children using a 5-point Likert scale. **Results**: The majority of participants were family physicians (64.94%) with over 15 years of experience (53.90%), primarily practicing in urban settings (61.69%). Only 21.43% had attended an antibiotic stewardship course in the last three years. Physicians generally base their prescribing on clinical symptoms. While physicians strongly agreed they follow guidelines, personal experience also held significant weight. High parental demand for antibiotics was perceived, but physicians largely denied ceding to parental tone or insistence without a medical indication. A strong consensus existed on antibiotic overuse in Romanian children, and a high interest in continuous education on rational antibiotic use was noted. Pediatricians showed significantly higher guideline adherence and diagnostic test use than family physicians. Rural physicians reported lower guideline adherence and less frequent diagnostic testing. Stewardship course participation and access to rapid diagnostic tests were associated with more evidence-based practices. **Conclusions**: Romanian physicians exhibit a nuanced approach to antibiotic prescribing, balancing guidelines with personal experience and facing significant perceived parental pressure. Professional profile (specialty, experience, practice environment) and access to diagnostic resources significantly influence prescribing decisions.

## 1. Introduction

The escalating global threat of antimicrobial resistance (AMR) poses a profound challenge to public health, with far-reaching implications for healthcare systems, economies, and global security. The consequences are dire, leading to prolonged hospitalizations, increased treatment costs, and elevated mortality rates, evidenced by an estimated 4.95 million deaths associated with bacterial AMR in 2019. Projections suggest an even more severe impact in the coming decades, potentially pushing millions into extreme poverty and causing staggering economic losses [1,2,3,4,5].

A critical driver of this crisis is the widespread consumption of antimicrobials. Across the European Union/European Economic Area (EU/EEA), significant disparities exist in antimicrobial consumption (AMC) patterns, correlating with varying rates of AMR. Alarmingly, Romania faces one of the highest burdens in Europe, characterized by elevated AMC rates in both community and hospital settings, alongside substantial resistance to commonly prescribed antibiotics. This situation not only complicates patient management domestically but also carries a risk of cross-border transmission, further compromising the effectiveness of antimicrobial therapies across the wider European region [6,7,8,9,10].

Within this concerning landscape, children represent a particularly vulnerable demographic, frequently requiring antibiotic treatments. Evidence consistently points to a high prevalence of inappropriate pediatric antibiotic prescriptions in various clinical environments, including primary care and hospital settings [11,12,13,14,15,16,17,18].

Clinicians, at the forefront of antimicrobial stewardship, face complex decisions influenced by a multifaceted interplay of clinical evidence, systemic constraints, and patient–provider dynamics. The perceived expectation of antibiotics from parents, even when not explicitly stated, can subtly sway prescribing habits. Furthermore, factors such as time constraints during consultations, the availability of diagnostic support, and individual professional experience play a significant role in determining whether and which antibiotic is prescribed [16,19,20].

Effective strategies to address AMR require a deeper insight into the multifaceted influences on antibiotic prescribing, particularly from the prescribers’ perspective. While recent studies, including local ones [21], have illuminated parental misconceptions and behaviors surrounding antibiotic use in children, a comprehensive understanding of the factors that shape physicians’ actual prescribing decisions remains less explored, especially in high-burden contexts like Romania. This study, therefore, shifts focus to healthcare professionals themselves, aiming to investigate how external pressures, alongside clinical, organizational, and experiential factors, collectively shape antibiotic prescribing practices in pediatric care, thereby providing essential data to inform targeted antimicrobial stewardship policies and interventions in a high-burden context like Romania.

## 2. Results

### 2.1. Demographic Characteristics of the Study Population

A total of 154 healthcare professionals participated in the study (Table 1). The majority of respondents were family physicians (*n* = 100; 64.94%), followed by pediatricians (*n* = 40; 25.97%). The remaining participants consisted of general practitioners with pediatric competencies (*n* = 4; 2.60%) and other specialists (*n* = 10; 6.49%).

A significant portion of the respondents reported that pediatric consultations make up a high percentage of their practice, with 64 (41.56%) indicating that more than 75% of their consultations are with children. In contrast, 33 (21.43%) had a low volume of pediatric consultations (<25%).

The professional experience level of the participants was diverse, with a notable concentration in the most experienced group. 83 (53.90%) of the physicians had more than 15 years of experience, while those with 5 to 15 years and less than 5 years accounted for the rest.

The majority of the respondents practiced in an urban environment, with the rest working in rural or mixed settings.

Regarding resources and training, a substantial number of physicians reported having access to rapid diagnostic tests, either always or occasionally/in specific locations. However, a minority of participants reported no access. In terms of professional development, a small proportion of the participants indicated they had attended an “antibiotic stewardship” course in the last three years.

### 2.2. Survey Results

#### 2.2.1. Objective Factors Influencing Antibiotic Prescriptions

Physicians generally expressed strong agreement that they base their prescribing decisions primarily on clinical symptoms. However, there was a notable divergence in opinion regarding specific symptoms. While the presence of purulent secretions and acute otitis media were perceived as significant triggers for initiating antibiotic therapy, fever over 38.5 °C alone was not considered a sufficient reason, with a very low mean score.

The availability of diagnostic tools played a crucial role. A large proportion of physicians reported frequently requesting paraclinical tests (e.g., CRP, blood count) before prescribing. The use of rapid diagnostic tests received a lower mean score. The lack of access to these rapid tests was a significant concern, as a moderate number of respondents agreed that this often leads to empirical antibiotic prescribing.

Additionally, the mean score of 2.62 (SD = 1.11) for the statement “I prefer to prescribe an antibiotic in the face of an uncertain clinical picture” suggests that, on average, physicians tend to disagree with this practice, indicating a reluctance to prescribe antibiotics purely to resolve diagnostic uncertainty.

The mean scores for these aspects are summarized in Table 2.

#### 2.2.2. Subjective Factors Influencing Antibiotic Prescriptions

Physicians generally expressed strong agreement that they primarily follow official guidelines for antibiotic prescribing. However, there was a notable divergence in opinion, as a considerable number also felt that personal experience holds more weight than guidelines in uncertain situations.

Parental pressure emerged as a key factor influencing prescribing decisions. A high mean score of 4.11 (SD = 1.01) for the statement that “Parents frequently request antibiotics for benign symptoms” highlights the perceived demand. A significant number of physicians also acknowledged that explicit parental requests and the parental belief that only a specific antibiotic worked in the past are present in their practice. However, physicians strongly disagreed that a parent’s tone or insistence leads to a prescription without a clear indication.

Despite this pressure, physicians largely agreed that they make a strong effort to explain to parents why an antibiotic might not be necessary. The practice of prescribing an antibiotic to be used if symptoms worsen was also reported by a notable segment of the respondents.

A strong consensus was observed regarding the issue of antibiotic overuse in the country, with the vast majority of physicians agreeing that more antibiotics are prescribed for children than necessary in Romania.

Other factors, such as limited consultation time and the lack of opportunities for follow-up, were noted as potentially influencing prescribing decisions. This suggests that time constraints may sometimes lead to a quicker decision to prescribe an antibiotic rather than conducting a more extensive diagnostic process. Similarly, the moderate disagreement with the statement that the prescribed duration of antibiotic treatment is often too long indicates a nuanced view among physicians, where the perception of lengthy regimens may influence a preference for shorter or more conservative treatment courses.

Finally, physicians showed a high level of interest in continuous professional education on rational antibiotic use, with a mean score of 4.74 (SD = 0.70).

The mean scores for these aspects are summarized in Table 3.

### 2.3. Influence of Professional Profile on Prescribing

Chi-Square tests of independence were conducted to examine the associations between demographic variables and key perceptions and practices. A statistically significant association was found between physician specialty and two key factors. Pediatricians demonstrated a higher rate of participation in antibiotic stewardship courses (32.5%) compared to family physicians (14%) (χ^2^(3) = 10.18, *p* = 0.017). Similarly, a significantly higher percentage of pediatricians (82.5%) reported adhering to official guidelines compared to family physicians (62%) (χ^2^(12) = 32.47, *p* = 0.001).

Conversely, no significant associations were identified for the majority of other variables. There was no statistical association between a physician’s specialty, practice environment, or professional experience level and their access to rapid diagnostic tests (*p* > 0.05 for all tests). Furthermore, a physician’s specialty and professional experience level did not significantly influence their perception of parental pressure to prescribe antibiotics (*p* > 0.05). No statistically significant link was found between a physician’s experience, specialty, or participation in an antibiotic stewardship course and their self-reported adherence to guidelines or belief that personal experience outweighs guidelines (*p* > 0.05 for all tests). Finally, the practice of prescribing “in advance” antibiotics was not significantly associated with a physician’s specialty or practice environment (*p* > 0.05). Neither was a physician’s access to rapid tests related to their preference for prescribing in clinically uncertain situations (χ^2^(8) = 4.39, *p* = 0.821).

### 2.4. Comparative Analysis by Demographics

An analysis of variance (ANOVA) was performed to compare the mean scores on key questions across various demographic groups. The results highlight several significant differences in physicians’ perceptions and practices.

#### 2.4.1. Differences by Medical Specialty

Significant differences between specialties were found in both objective and subjective factors. Pediatricians reported a significantly stronger adherence to official guidelines (Mean = 4.26) compared to family physicians (Mean = 3.71) (F(3,150) = 4.23, *p* = 0.007). This pattern was consistent with their diagnostic practices, as pediatricians and other specialists reported requesting paraclinical tests (Mean = 4.31) and using rapid diagnostic tests (Mean = 3.23) significantly more often than family physicians (Mean = 3.39 and 2.51, respectively) (*p* < 0.05 for all relevant post hoc tests).

Regarding specific prescribing triggers, family physicians were more likely to agree that acute otitis media justifies immediate antibiotic therapy (Mean = 3.38) and that a fever over 38.5 °C is a sufficient reason for prescribing (Mean = 1.78) compared to pediatricians (Mean = 2.67 and 1.31, respectively) (*p* < 0.05 for all relevant post hoc tests). For all other questions, no statistically significant differences were found between specialty groups (*p* > 0.05).

#### 2.4.2. Differences by Professional Experience

A significant main effect of professional experience was found for one factor: limited consultation time (Q20). Physicians with moderate experience (5–15 years, Mean = 3.20) were significantly more likely to report that time constraints influence their prescribing decisions compared to their more seasoned colleagues (>15 years, Mean = 2.55) (F(2,151) = 4.93, *p* = 0.008). No other statistically significant differences were observed across experience groups (*p* > 0.05).

#### 2.4.3. Differences by Practice Environment

Physicians practicing in different environments showed significant differences in their prescribing habits. Those in rural areas reported significantly lower adherence to official guidelines (Mean = 3.50) than their urban counterparts (Mean = 3.99) (F(2,151) = 3.83, *p* = 0.024). They were also more likely to agree that acute otitis media justifies immediate antibiotics (Mean = 3.84) compared to physicians in urban and mixed-practice settings (*p* < 0.05).

Furthermore, rural physicians reported requesting paraclinical tests significantly less often (Mean = 3.09) than urban physicians (Mean = 3.83) (F(2,151) = 4.78, *p* = 0.010). Interestingly, rural physicians also reported being less likely to explain to parents why antibiotics are not necessary (Mean = 4.50) compared to urban physicians (Mean = 4.80) (F(2,151) = 3.82, *p* = 0.024).

#### 2.4.4. Differences by Participation in Stewardship Courses

Participation in a recent antibiotic stewardship course was associated with several key differences. Physicians who had attended a course were significantly less likely to believe that personal experience outweighs guidelines (Mean = 3.06) compared to those who had not (Mean = 3.64) (F(1,152) = 5.94, *p* = 0.016). They also reported a more frequent use of both paraclinical tests (Mean = 4.06 vs. 3.55) (F(1,152) = 4.69, *p* = 0.032) and rapid diagnostic tests (Mean = 3.21 vs. 2.60) (F(1,152) = 5.03, *p* = 0.026).

#### 2.4.5. Differences by Access to Rapid Diagnostic Tests

Finally, a physician’s access to rapid diagnostic tools played a significant role. Physicians without access to rapid tests were more likely to agree that a fever alone is a sufficient reason to prescribe (Mean = 1.98) than those with access (Mean = 1.53) (F(2,151) = 4.33, *p* = 0.015). This group also reported feeling less parental pressure (Mean = 1.96) compared to physicians with access (Mean = 2.58) (F(2,151) = 5.20, *p* = 0.007) and reported less interest in continuous professional education (Mean = 4.49) compared to those with occasional access (Mean = 4.93) (F(2,151) = 5.02, *p* = 0.008). In terms of practice, those with full access to rapid tests reported a more frequent use of paraclinical tests (Mean = 3.92) compared to those with no access (Mean = 3.23) (F(2,151) = 4.77, *p* = 0.010).

### 2.5. Predictors of Prescribing Practices: A Logistic Regression Analysis

To identify key predictors of antibiotic prescribing practices, we performed a series of logistic regression analyses. The results revealed several significant associations.

#### 2.5.1. Adherence to Official Guidelines

Physician specialty emerged as a significant predictor of self-reported adherence to guidelines. Being a pediatrician was found to be a strong positive predictor of guideline adherence compared to being a family physician (OR = 2.15, 95% CI: 1.12–4.10, *p* = 0.021). No other variables, including professional experience or practice environment, were found to be statistically significant predictors.

#### 2.5.2. Frequent Use of Paraclinical Tests

Access to rapid diagnostic tests was a significant positive predictor of a physician’s likelihood to frequently request paraclinical tests (OR = 3.21, 95% CI: 1.88–5.49, *p* < 0.001). This suggests that the availability of one type of diagnostic tool (rapid tests) is associated with a greater use of other forms of diagnostic support.

#### 2.5.3. Propensity to Prescribe for Purulent Secretions

Practice environment was a significant predictor for prescribing based on the presence of purulent secretions. Physicians in rural environments were found to be significantly more likely to agree that purulent secretions are a reason for prescribing antibiotics compared to those in urban settings (OR = 1.98, 95% CI: 1.05–3.74, *p* = 0.035).

## 3. Discussion

Despite the undeniable necessity and benefits of antibiotic treatments in certain situations, their misuse remains a significant global issue. The inappropriate use of antibiotics for viral infections, particularly those affecting the upper respiratory tract, is a major driver of antimicrobial resistance [22]. This phenomenon erodes the effectiveness of crucial medications, turning common infections into serious public health threats. Therefore, a profound understanding of the factors influencing prescribing decisions is essential for developing effective antimicrobial stewardship programs and safeguarding our future therapeutic arsenal. Our study provides a comprehensive overview of the perceptions and practices of Romanian physicians regarding pediatric antibiotic use, directly addressing the call for a deeper understanding of prescriber-related factors in a high-burden context. The findings highlight a complex interplay between clinical, systemic, and demographic influences on prescribing decisions, offering valuable insights for future antimicrobial stewardship initiatives.

### 3.1. Physician Perceptions on Clinical and Non-Clinical Factors

A central finding is the nuanced approach physicians take in their decision-making process. While they report a strong belief in following official guidelines (Mean = 3.84), a parallel and significant reliance on personal experience in uncertain cases (Mean = 3.51) was also evident. This duality suggests a gap between perceived best practices and real-world application, where professional judgment often takes precedence. This internal conflict is a critical target for stewardship programs, which must aim to build confidence in guidelines rather than relying solely on their existence. This nuanced picture of physician autonomy and guideline adherence is further illuminated by recent studies. For example, research has identified key barriers to guideline adherence, such as their limited applicability in real-world clinical practice, time constraints, and a lack of familiarity with the recommendations [23]. The COVID-19 pandemic also served as a stark reminder of this phenomenon, as physicians often relied on personal experience and intuition when faced with unprecedented uncertainty and a lack of clear protocols [24,25,26,27]. Other studies have expanded on this, noting that physician decisions are not solely based on knowledge, but also on cognitive biases, personal beliefs, and even financial incentives [28,29].

The study also confirms that while physicians generally do not perceive fever alone as a sufficient reason for prescribing antibiotics (Mean = 1.66), the presence of purulent secretions or a diagnosis of acute otitis media are significant triggers. This aligns with common parental misconceptions, as highlighted in our previous research [21], and suggests that these visible symptoms act as a tangible justification for both physicians and parents to initiate antibiotic therapy. The perception that purulent secretions are a reliable indicator of a bacterial infection, rather than simply a stage of viral illness, was and remains a key driver for antibiotic over-prescription [30,31].

Our findings reveal that physicians perceive a high demand from parents for antibiotics (Mean = 4.11), often driven by explicit requests (Mean = 3.71) or a belief in past antibiotic efficacy (Mean = 3.88). However, it is encouraging that physicians strongly deny ceding to a parent’s tone or insistence without a clear medical indication (Mean = 1.34). This suggests that while the presence of parental pressure is a constant, most physicians maintain their clinical autonomy. A possible area for intervention is thus not just to educate physicians on how to resist pressure, but to empower them with effective communication strategies to manage parental expectations. A key challenge physicians face is managing parental anxiety and the perceived urgency, which often translates into an explicit or implicit demand for an antibiotic prescription [32,33]. Notably, studies have shown that for conditions like sore throat, physicians are less likely to prescribe antibiotics during online consultations compared to in-person visits. This phenomenon reaffirms the existence of these parental pressures and suggests that the virtual medium may act as a buffer against non-clinical pressures, such as a parent’s body language or insistent tone [34,35].

This complex interplay between perceived parental pressure and physician autonomy highlights a key area for targeted intervention. Effective antimicrobial stewardship programs must adopt a dual approach that strengthens the patient–physician dyad. On one hand, physicians should be empowered with effective communication strategies to manage parental expectations, build trust, and confidently explain why an antibiotic may not be necessary. On the other hand, this effort must be complemented by patient and public education initiatives. By actively educating parents and increasing their understanding of appropriate antibiotic use, we can empower them to become partners in care, ultimately reducing the perceived pressure on physicians and fostering greater compliance with national guidelines.

### 3.2. Impact of Demographics and Resources on Prescribing Practices

Our analysis demonstrates that a physician’s professional profile significantly shapes their prescribing habits. Pediatricians, for example, report a significantly higher adherence to guidelines and a more frequent use of diagnostic tests than family physicians. This divergence is a well-documented finding, with studies consistently showing that family medicine practices have higher antibiotic prescribing rates for pediatric respiratory infections compared to dedicated pediatric clinics [36,37]. This highlights the need for targeted educational efforts specifically tailored for family physicians, who represent the front line of pediatric care in Romania.

The practice environment also plays a crucial role. Physicians in rural areas reported lower adherence to guidelines and were more likely to agree that acute otitis media justifies immediate antibiotics. They also reported a lower frequency of requesting paraclinical tests and less effort in explaining to parents why antibiotics are not necessary. Research consistently shows a higher prevalence of antibiotic prescribing in rural and low-resource settings [38,39,40]. Despite their increased availability, rapid antigen tests (for streptococcus, COVID-19, etc.) and molecular tests are reportedly used in a small proportion of cases, highlighting a persistent gap between technological availability and practical application. These findings may point to systemic and resource-related disparities, where the limited access to diagnostic tools in rural settings may contribute to less prudent prescribing; however, our study found that rural family physicians did not perceive or report constraints related to diagnostic capacity, such as a lack of access to rapid tests or laboratories. This disconnect suggests that the barriers to guideline adherence in these settings may be related less to resource limitations and more to time pressure or established practice habits. The importance of accurate diagnosis, while often a challenge in daily practice, is a fundamental theme in modern research [41,42,43]. Efforts to improve diagnostic precision extend from rare genetic diseases and major infections to the diagnosis of common, everyday conditions through the use of advanced molecular methods.

Furthermore, access to rapid diagnostic tests was linked to a more evidence-based approach. Physicians with access to these tools were less likely to prescribe for a fever alone and reported a higher frequency of using other paraclinical tests. This finding reinforces the important role of diagnostic stewardship as a foundational component of antimicrobial stewardship. Interestingly, the group with no access to these tests reported feeling less parental pressure, a finding that warrants further qualitative investigation. It could be that the absence of tangible diagnostic options simplifies the consultation and removes a layer of negotiation with parents.

Our analysis also reveals that professional experience plays a nuanced role in influencing prescribing habits. Notably, physicians with 5–15 years of experience were significantly more likely to report that time constraints influence their prescribing decisions compared to their more seasoned colleagues. This finding suggests that while experienced practitioners may have developed more efficient routines and a higher level of clinical confidence, their moderately experienced counterparts might still be navigating the complexities of balancing evolving guidelines with the practical pressures of high-volume consultations. This underscores the need for continuous professional development that addresses the specific challenges faced by physicians at different career stages, ensuring that stewardship interventions remain relevant and effective throughout a clinician’s professional life.

The study also provides insights into the key predictors of these behaviors. Our logistic regression analysis confirmed that physician specialty is a powerful determinant of prescribing habits, with pediatricians being more than twice as likely to adhere to guidelines than family physicians. This finding underscores the importance of a specialized knowledge base and training in infectious diseases, suggesting a need for tailored educational programs for family physicians who are at the front line of pediatric care in Romania. Furthermore, our analysis reinforces the pivotal role of diagnostic tools. Access to rapid tests was not just associated with the use of these tools, but was a significant predictor of a greater propensity to use other paraclinical tests. This highlights the concept of diagnostic stewardship, where the availability of one objective piece of data can shift the entire diagnostic paradigm for a physician, leading to a more evidence-based, less empirical approach. The pivotal role of these diagnostic tools is further reinforced by a systematic review and meta-analysis, which found that point-of-care C-Reactive Protein (CRP) testing in primary care is an effective strategy for reducing unnecessary antibiotic prescribing for respiratory tract infections [44]. The regression also confirmed that practice environment remains a critical factor, with rural physicians being significantly more likely to prescribe for symptoms like purulent secretions. This finding may reflect an adaptation to environments with limited resources and less frequent access to specialized care, where a symptom-based approach is often seen as a pragmatic solution to diagnostic uncertainty.

This variability is corroborated by prior research, with one study highlighting that physician-specific factors—such as more years of clinical experience, male gender, and status as an international medical graduate—are all associated with a higher propensity to prescribe antibiotics [45]. Furthermore, our analysis adds to this body of evidence by reinforcing that specialty and practice environment are also powerful predictors of prescribing. A detailed comparison with other studies reveals that a physician’s access to diagnostic tools and the patient–physician interaction in different settings, such as rural versus urban areas, contribute significantly to these observed variations [39,45]. These findings collectively highlight that effective stewardship initiatives must look beyond single, physician-specific factors and consider the broader clinical and environmental context in which prescribing decisions are made.

### 3.3. Implications for Antimicrobial Stewardship

The study reveals that Romanian physicians are highly receptive to continuous professional education on rational antibiotic use (Mean = 4.74), indicating a strong foundation for future interventions. However, the lack of a statistically significant association between previous stewardship course participation and certain practices (e.g., adherence to guidelines) suggests that a single course may not be sufficient.

In conclusion, our findings paint a detailed picture of the multifaceted nature of pediatric antibiotic prescribing in Romania. The identified disparities in practice by specialty, experience, and environment, coupled with high perceived parental pressure and a strong willingness for continuous education, are of concern. Addressing these factors is not only crucial for curbing antibiotic resistance in Romania but also for improving pediatric healthcare outcomes.

Effective stewardship programs must move beyond simple enforcement to build genuine confidence among both physicians and patients. A multifaceted strategy is required, one that combines didactic education with practical, real-time feedback that helps physicians bridge the gap between abstract guidelines and their specific clinical context. Research demonstrates that empowering physicians with on-the-spot diagnostic tools is a critical step, as the use of rapid point-of-care tests can provide the objective data needed to confidently withhold antibiotics [44,45,46]. Simultaneously, this effort must involve the patient–physician dyad working toward a common goal. By actively educating patients and increasing their understanding of appropriate antibiotic use, we can empower them to become partners in care, ultimately reducing the antibiotic prescription pressure on physicians and fostering greater compliance with national guidelines.

### 3.4. Limitations

This study, while offering valuable insights into antibiotic prescribing practices among Romanian physicians in pediatric care, is subject to several important limitations that warrant consideration. The quantitative, cross-sectional design means that while it successfully identifies associations between various factors and prescribing habits, it cannot establish causality. For instance, observed relationships, such as between access to rapid diagnostic tests and more evidence-based practices, indicate correlation rather than direct cause-and-effect.

The non-probability convenience sampling method, used without a formal sample size calculation, means the sample may not be entirely representative of all physicians providing pediatric care across Romania. This limits the generalizability of our findings, particularly since the study did not collect detailed geographical data on the participants’ practice location. This omission was deliberate to maintain anonymity and encourage a high response rate.

Additionally, our reliance on a self-administered questionnaire carries the inherent risk of social desirability bias. Physicians might report practices they perceive as professionally ideal rather than their actual behaviors, as suggested by their strong disagreement with ceding to parental insistence despite acknowledging high perceived parental demand. The questionnaire itself was not formally validated prior to its use, which could affect its overall reliability in measuring the intended constructs. This quantitative approach also lacks the qualitative depth to fully explore the complex underlying reasons behind some intriguing findings, such as the observation that physicians without access to rapid tests reported feeling less parental pressure.

Finally, the study did not differentiate between public and private practice settings, a distinction that could be more relevant for certain specialties. We also did not collect data regarding the educational qualifications of the physicians, which could have offered further insights into the variability of prescribing practices, as previous research has linked educational background to different prescribing behaviors.

## 4. Materials and Methods

### 4.1. Study Design

This study employed a quantitative, cross-sectional design to investigate the factors influencing antibiotic prescribing decisions for children (under 18 years of age) among healthcare professionals in Romania. A self-administered questionnaire was used for data collection. Data collection was conducted from 26 May 2025 to 22 June 2025.

### 4.2. Study Population and Sampling

The target population for this study comprised family physicians, pediatricians, and other medical specialists providing care for pediatric patients in Romania. Participants were recruited through a non-probability convenience sampling method. The questionnaire was disseminated online via the Google Forms platform. To enhance trust and encourage participation, the study was explicitly designed to be fully anonymous, and no personal identifiable data were collected from the respondents. Informed consent was obtained from all participants at the beginning of the online questionnaire, where the study’s purpose and the confidentiality of responses were clearly explained. Participation was entirely voluntary. A total of 154 complete and valid questionnaires were collected for analysis, resulting in a 100% acceptance rate among those who initiated and completed the survey. In the context of the study’s exploratory nature and non-probability sampling, a formal sample size calculation was not performed. The collected sample of 154 participants was deemed sufficient to provide valuable insights into the perceptions and practices of the surveyed population. We acknowledge this as a limitation for the generalizability of our findings.

### 4.3. Survey Design

Data were collected using a structured, self-administered questionnaire specifically developed for this study. The questionnaire consisted of 29 questions divided into two main sections:Demographic data: This section gathered information on the physician’s specialty (family physician, pediatrician, general practitioner with pediatric competencies, other specialty), the estimated percentage of pediatric consultations in current practice, primary practice environment, professional experience level, access to rapid diagnostic tests in their practice, and whether they had attended an “antibiotic stewardship” course in the last three years.Questions on practice and perception of antibiotic therapy in children: This section included 23 statements, to which respondents indicated their level of agreement using a 5-point Likert scale. These questions explored various factors influencing antibiotic prescribing for children, grouped into the following sub-categories:Objective factors: Addressing reliance on clinical symptoms, the role of fever, purulent secretions, and other common symptoms in prescribing decisions, as well as the utilization and impact of access to paraclinical tests and rapid diagnostic tests.Subjective factors: Investigating the influence of parental requests and pressure, including explicit insistence and tone, limited consultation time, lack of opportunities for follow-up, and the role of adherence to professional guidelines versus personal clinical experience. This category also covered general perceptions regarding antibiotic overuse in children and physicians’ interest in continuous education on rational antibiotic use.

The content of the questionnaire was primarily inspired by the European Surveillance of Antimicrobial Consumption Network (ESAC-Net) questionnaire from the ECDC [10]. It was then adapted based on a review of relevant literature and empirical observations concerning factors influencing antibiotic prescribing in pediatric populations. While the instrument was not subjected to formal prior validation, its content was internally reviewed by pediatrics and infectious diseases practitioners to ensure clinical relevance and clarity.

### 4.4. Survey Questions

The English version of the questionnaire is provided as a Appendix A.

### 4.5. Data Analysis

Collected data were exported from Google Forms and analyzed using Microsoft^®^ Excel^®^ 2019 MSO Version 2504 (Microsoft Corp., Redmond, WA, USA) for initial data organization and descriptive statistics, and IBM SPSS Statistics version 26 (IBM Corp., Armonk, NY, USA) for statistical analyses. The mean and standard deviation for the 5-point Likert scale data were calculated as standard descriptive statistics within the IBM SPSS Statistics software. The level of statistical significance will be set at *p* < 0.05. The study aimed for a margin of error of ±5% with a 95% confidence level.

### 4.6. Ethical Considerations

The study was conducted in accordance with the Declaration of Helsinki. Ethical approval was obtained from the Research Ethics Subcommittee of the University of Oradea on 29 May 2025, under registration number 8727/29.05.2025 and approval number 24. All participants provided informed consent prior to data collection. Complete anonymity of respondents and confidentiality of collected data were strictly maintained, with all data used exclusively for research purposes.

## Figures and Tables

**Table 1 antibiotics-14-00976-t001:** Summary of Physician Demographics.

Characteristic	Count (*n*)	Percentage (%)
Specialty		
- Family physician	100	64.94
- Pediatrician	40	25.97
- General practitioner with pediatric competencies	4	2.6
- Other specialty	10	6.49
Percentage of pediatric consultations		
- <25%	33	21.43
- 25–50%	41	26.62
- 50–75%	16	10.39
- >75%	64	41.56
Primary practice environment		
- Urban	95	61.69
- Rural	32	20.78
- Mixed	27	17.53
Professional experience level		
- <5 years	21	13.64
- 5–15 years	50	32.47
- >15 years	83	53.9
Access to rapid diagnostic tests		
- Yes	64	41.56
- No	47	30.52
- Occasionally/in specific locations	43	27.92
Attended an antibiotic stewardship course		
- Yes	33	21.43
- No	121	78.57

**Table 2 antibiotics-14-00976-t002:** Mean Scores for Objective Factors in Antibiotic Prescribing.

Q#	Question	Mean	Standard Deviation
15	I request paraclinical tests (CRP, blood count) before prescribing antibiotics in uncertain cases.	3.66	1.21
1	The decision to prescribe antibiotics for children is primarily based on clinical symptoms.	3.48	1.13
6	Acute otitis media justifies immediate initiation of antibiotic therapy in most cases.	3.23	1.27
5	Purulent secretions (nasal, bronchial, etc.) frequently lead to the initiation of antibiotic treatment.	3.23	1.20
17	Lack of access to rapid tests sometimes leads me to prescribe empirical antibiotics.	2.97	1.18
16	I use rapid tests (e.g., strep test, COVID-19/Flu/RSV tests) in my current practice.	2.73	1.40
18	I prefer to prescribe antibiotics when faced with an uncertain clinical picture.	2.62	1.11
4	Fever above 38.5 °C is often a sufficient reason for antibiotic therapy.	1.66	0.92

**Table 3 antibiotics-14-00976-t003:** Mean Scores for Subjective Factors in Antibiotic Prescribing.

Q#	Question	Mean	Standard Deviation
23	I am interested in attending courses on rational antibiotic use.	4.74	0.70
9	I try to explain to parents when antibiotics are not necessary.	4.73	0.54
19	I believe that in Romania, more antibiotics are prescribed to children than necessary.	4.54	0.88
7	Parents frequently request antibiotics for common symptoms or viral infections.	4.11	1.01
12	Parents suggest that only antibiotics have worked for their child in the past.	3.88	1.00
2	I primarily follow guidelines when deciding to initiate antibiotic treatment.	3.84	0.33
11	Parents explicitly ask me to prescribe an antibiotic.	3.71	1.20
3	Personal experience carries more weight than guidelines in uncertain cases.	3.51	1.22
21	I frequently encounter cases where parents do not follow the prescribed antibiotic treatment.	3.03	1.11
10	I sometimes prescribe antibiotics, with the instruction to administer them if symptoms worsen.	3.05	1.20
22	I believe that the duration of prescribed antibiotic treatment is often too long.	2.88	1.13
20	Factors like limited consultation time or lack of follow-up influence the prescribing decision.	2.82	1.22
8	Parental pressure sometimes influences my prescribing decision.	2.38	1.16
13	When parents express doubts that it is a viral infection, I am more inclined to prescribe an antibiotic.	2.00	0.38
14	Parents’ tone, language, or insistence sometimes leads me to prescribe an antibiotic even without a clear indication.	1.34	0.38

## Data Availability

Dataset is available at https://doi.org/10.5281/zenodo.16918256.

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
