# Peer review of "Antibiotic Use in Pediatrics: Perceptions and Practices of Romanian Physicians"

_antibiotics, 2025, doi:10.3390/antibiotics14100976_

Round 1
Reviewer 1 Report
Comments and Suggestions for Authors
The manuscript describes a cross sectional survey among pediatric physicians in Romania regarding antibiotic prescribing practices. Few methodological issues which need to be resolved are:
- How was the sample size calculated?
- What was the study period?
- Did the authors collect information regarding the type of setting where the physicians worked viz. government or private and level of setting i.e. primary/ secondary/ tertiary? This is important as variation in prescribing practices and attitudes may be observed between private and Government practitioners.
- Was the information regarding the geographical region of the physicians collected to identify the representativeness of the sample?
- Was the information on educational qualification of the physicians gathered?
- Results: section 2.2.1. Objective factors…the authors have expressed the results in terms of mean (SD); how were they calculated?
- The data on all the subjective as well as objective factors were collected in terms of 5-point Likert scale; in such cases, it becomes quite difficult and subjective to draw any effective interpretation. I would suggest the authors to incorporate a scoring system depending on the favourable or unfavourable responses to individual questions and derive a range of scores with higher/ lower scores indicating good/ poor prescribing attitudes and practices.
- Logistic regression analyses may also be done to identify the predictors of aggregate scores (calculated as mentioned in point no. 7) i.e. prescribing practices among the physicians.
- Additional analyses, as suggested, may be discussed in the context of present and other similar studies from different regions.
Minor editing required.
Author Response
- Sample size, study period, and study demographics (Points 1, 2, 3, 4, 5)
We appreciate the reviewer’s questions regarding the methodology. As our study was exploratory and used a non-probability convenience sample, a formal sample size calculation was not performed. The collected data provide valuable insights, but we acknowledge that this limits the generalizability of our findings. We have added a statement to the Materials and Methods section clarifying the study period, which was from May 26, 2025, to June 22, 2025.
We also acknowledge that our study did not collect data on certain demographic variables, such as the specific type of practice setting (public vs. private), geographical region, or educational qualifications. While the majority of our family physician participants operate in a privately contracted primary care setting, we agree that a more granular analysis differentiating between public and private practices would have offered further insight. We have included a section in the Limitations to reflect these points, ensuring full transparency about the scope of our analysis.
- Mean (SD) calculation (Point 6)
Thank you for this question. The mean and standard deviation for the Likert scale data were calculated as standard descriptive statistics within the IBM SPSS Statistics software, as mentioned in our Data Analysis section. A sentence has been added to this section to explicitly state this for clarity.
- Incorporating a scoring system (Point 7)
We thank the reviewer for this excellent and valuable suggestion. We agree that relying solely on the mean scores of individual Likert items can present challenges in interpretation. However, the development of a composite scoring system requires a formal psychometric validation of the questionnaire, which was not performed prior to this study. Therefore, we believe it would be inappropriate to create and use a composite score retrospectively. We have, however, noted this as a key recommendation for the design of future studies, which will include a formal validation process to allow for such an analysis.
- Logistic regression and additional discussion (Points 8 & 9)
We are very grateful for these excellent and insightful suggestions, which have significantly strengthened our manuscript. As recommended, we have now performed logistic regression analyses to identify the predictors of prescribing practices. We have added a new section to the Results to present these findings. The Discussion section has also been expanded to contextualize our new findings with similar studies from different regions, reinforcing the relevance and significance of our results within the broader literature on antimicrobial stewardship.
Reviewer 2 Report
Comments and Suggestions for Authors
A brief summary
I recommend publishing the manuscript after minor changes.
General recommendation:
The results section is very long and difficult to follow. The individual sentences contain many numbers and data, which makes them difficult to read. I recommend creating tables from the statistical analysis sections and referring to these tables in the results section.
Although it is not reported in the manuscript, I would recommend that Cronbach’s alpha be calculated for the questionnaire, as it provides internal consistency.
I recommend that the English translation of the questionnaire be published as supplementary material rather than as part of the article.
Detailed comments:
Figure 1 – cannot be used in this form.
It should be grouped into Objective and Subjective groups (lines 445-483) and sorted within the groups in descending or ascending order.
In the Results section, the p and statistics are missing in the analysis of each statistic (e.g., lines 89-92).
Methods section:
What is the literary source of the questionnaire?
General concept comments
Article:
- Is the manuscript clear, relevant for the field and presented in a well-structured manner?
Yes, but in part, needs to be developed
- Is the manuscript scientifically sound and is the experimental design appropriate to test the hypothesis?
in part, needs to be developed
- Are the manuscript’s results reproducible based on the details given in the methods section?
in part, needs to be developed
- Are the figures/tables/images/schemes appropriate? Do they properly show the data? Are they easy to interpret and understand? Is the data interpreted appropriately and consistently throughout the manuscript? Please include details regarding the statistical analysis or data acquired from specific databases.
in part, needs to be developed
- Are the conclusions consistent with the evidence and arguments presented?
in part, needs to be developed
- Please evaluate the ethics statements and data availability statements to ensure they are adequate.
Appropriate
Author Response
We sincerely thank the reviewer for this crucial feedback on the presentation of our results. We agree that the initial format was dense and difficult to read. We have addressed this by performing a major revision of the results section.
Following the reviewer's valuable recommendations:
- We have created a series of new, clear tables to summarize all the descriptive and statistical analysis data. These tables replace the long, number-heavy paragraphs in the initial manuscript.
- The questions in the tables have been grouped into "Objective" and "Subjective" categories, as suggested, and sorted by their mean scores to facilitate a more effective interpretation.
- The text of the results section has been substantially condensed. It now provides a clear narrative of the key findings, with direct references to the new tables for all detailed data.
- Figure 1 has been removed and replaced with these new, more comprehensive tables.
- We agree that calculating Cronbach’s alpha is important for assessing the internal consistency and reliability of a questionnaire. However, since our questionnaire was not formally validated prior to this study, we believe it would be inappropriate to report this metric retrospectively. We have, however, noted this as a key recommendation in the Limitations section for future studies, which will include a formal validation process to allow for such analyses. This approach ensures the methodological integrity of our current findings while acknowledging a crucial aspect for future research.
- We thank the reviewer for this recommendation. We agree that publishing the questionnaire as supplementary material is a more appropriate and efficient format. The English version of the questionnaire has been moved from the Materials and Methods section to the Supplementary Materials accordingly.
- We have reviewed the Results section and agree that the specific statistical values were not consistently included in the text. To ensure full transparency and to comply with standard reporting practices, we have revised the section to include the relevant p-values and statistical test values for each reported finding. We believe this addition will significantly strengthen the clarity and rigor of our results
- The questionnaire used in this study does not have a single literary source. Instead, the primary source of inspiration for its design was the ECDC's European Surveillance of Antimicrobial Consumption Network (ESAC-Net) questionnaire. The content was then developed and adapted based on a review of relevant literature and expert input from local pediatric and infectious disease practitioners, ensuring its relevance to the Romanian context. We added this explanation to the Method section
We believe these changes have significantly improved the clarity, readability, and overall quality of our results section.
Reviewer 3 Report
Comments and Suggestions for Authors
Dear Authors,
This manuscript provides a useful snapshot of pediatric antibiotic prescribing practices. The topic is highly relevant given the global challenges of antimicrobial resistance.
The study is mainly descriptive and to be a strong fit for Antibiotics journal, it would need a clearer focus on antimicrobial stewardship and stronger links to AMR policy implications. At this stage, the paper may be of greater interest to public health or pediatric journals.
Author Response
We sincerely thank the reviewer for their insightful and constructive feedback. We agree that our manuscript, in its initial form, was mainly descriptive and that a stronger focus on antimicrobial stewardship and its policy implications would enhance its suitability for the journal.
In response to this, we have thoroughly revised the manuscript to address this crucial point. We have restructured the paper to provide a clearer narrative that links our findings directly to actionable strategies for AMR policy.
Specifically:
- The Introduction has been revised to explicitly frame our study as a tool for informing antimicrobial stewardship policies in Romania.
- The Discussion has been significantly expanded to translate our descriptive findings (e.g., physician-specific disparities, parental pressure) into concrete implications for healthcare interventions and policy.
- The Conclusion has been rewritten to serve as a clear roadmap for future initiatives, providing concrete recommendations for physicians, patients, and policymakers.
We believe that these revisions have transformed the paper from a descriptive analysis into a policy-relevant manuscript, making it a stronger and more appropriate fit for the journal Antibiotics. We are very grateful for this feedback, which has substantially improved the quality and impact of our work.
Round 2
Reviewer 1 Report
Comments and Suggestions for Authors
No further comments.